# X-INSTRUCTBLIP: A FRAMEWORK FOR ALIGNING X-MODAL INSTRUCTION-AWARE REPRESENTATIONS TO LLMS AND EMERGENT CROSS-MODAL REASONING

## ABSTRACT

Vision-language pre-training and instruction tuning have demonstrated general-purpose capabilities in 2D visual reasoning tasks by aligning visual encoders with state-of-the-art large language models (LLM). In this paper, we introduce a simple, yet effective, multimodal framework built atop a frozen LLM; this framework is capable of seamlessly integrating and managing an ad-hoc number of modalities. To facilitate general-modality training, we collect high-quality instruction tuning data in an automatic and scalable manner, composed of 31K QA samples for audio and 250K QA samples for 3D. We further contribute a novel Discriminative Cross-modal Reasoning (DisCRn) evaluation task, comprising 12K audio-video QA samples and 28K image-3D QA samples. Leveraging instruction-aware representations, our model consistently outperforms or matches the leading-edge counterparts, setting state-of-the-art benchmarks in seven (7) zero-shot scenarios across all investigated modalities. Notably, our approach demonstrates joint reasoning abilities on par with models specifically trained on combined-modality datasets, like video-audio. All associated resources, including codes, datasets, and benchmarks, will be released.

## 1 INTRODUCTION

Humans inherently utilize multiple senses to interpret their surroundings and formulate decisions. By equipping artificial agents with the capability for cross-modal reasoning,[1] we can foster the development of systems with a more comprehensive understanding of their environment, allowing them to discern patterns and make inferences that are not apparent when analyzing modalities separately. This aspiration has motivated the advent of Multimodal Language Models (MLMs) (Alayrac et al., 2022; Huang et al., 2023; Gong et al., 2023; Li et al., 2023b; Koh et al., 2023; Dai et al., 2023; Liu et al., 2023; Driess et al., 2023b), which transfer the remarkable abilities of Large Language Models (LLMs) (Devlin et al., 2018; Brown et al., 2020; Raffel et al., 2020; Rae et al., 2021; Chowdhery et al., 2022; Tay et al., 2022; Chung et al., 2022; Touvron et al., 2023; Chiang et al., 2023; Taori et al., 2023) to the visual domain.

Recent advancements seek to extend the reasoning capabilities of models beyond static vision, by incorporating audio and video, either by introducing pre-trained foundation models across multiple modalities (Lu et al., 2022; Shukor et al., 2023; Xu et al., 2023a; Wang et al., 2023; Chen et al., 2023a;b) or by training projection modules to align multimodalities to the representation space of LLMs (Zhao et al., 2023; Wu et al., 2023). Although effective, the former necessitates task-specific fine-tuning for successful multimodal task execution, and the latter refines models on joint-modality data to carry out tasks involving combined modalities. Such data can be high-resource demanding both in terms of collection and computational resources. Several studies (Su et al., 2023; Han et al., 2023) have employed multimodal representation spaces (Girdhar et al., 2023) to extend vision-centric MLMs to additional modalities, but such frameworks are constrained only to the modalities facilitated by those representation spaces.

---

[1] In the scope of this work, the term *cross-modal reasoning* encompasses the ability of a system to process information from multiple modalities (excluding text) to perform tasks that require both integrating and discriminating information across those modalities. This term is chosen to distinguish from "multimodal tasks," a term traditionally reserved for tasks that involve the dual modalities of vision and language.

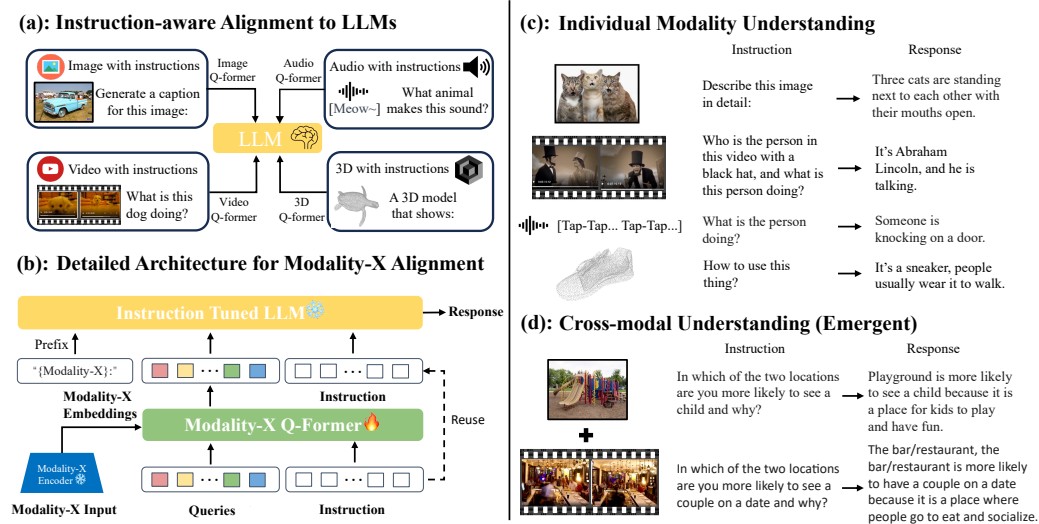

Figure 1: **(a)** `X-InstructBLIP` employs individually encoded modalities, each transformed to the language domain via a uniquely trained, instruction-aware Q-Former. **(b)** The input to the LLM is the Q-Former output representation, cued with a modality-specific prefix, `"Modality-X"` and followed by the instruction. This approach results in competitive understanding within individual modalities **(c)**) while revealing emergent capabilities for cross-modal comprehension (**(d)**).

In this study, we present `X-InstructBLIP`, a scalable and extendable framework, illustrated in Figure 1, developed to enable learning with uni-modal data without the constraints imposed by pre-trained universal cross-modal spaces or the computational resources and potential overfitting risks associated with unfreezing the parameters of the LLMs. `X-InstructBLIP` is designed to seamlessly incorporate a multitude of modalities, on an ad-hoc basis, eliminating the necessity for joint modality datasets but preserving the capacity to execute cross-modality tasks. Utilizing the Querying Transformer or Q-Former (Li et al., 2023b) module, we effectively map inputs from the separate embedding spaces attributed to each modality into the domain of LLMs. Considering the scarcity of instruction tuning data for a spectrum of modalities, we introduce a simplified yet potent approach: a three-stage query data augmentation technique. This strategy leverages open-source language models to extract instruction-tuning data from captioning datasets.

Our approach matches or surpasses the performance of existing models in unimodal reasoning benchmarks, covering seven zero-shot tasks across four modalities. Our model also demonstrates parallel competence in cross-modality reasoning compared to models trained with cross-modal data. To demonstrate this, we introduce `DisCRn`, an automatically curated **Dis**criminatory **C**ross-modal **R**easo**n**ing challenge dataset, which requires models to distinguish between diverse combinations of modalities, such as audio-video and 3D-image. A compilation of illustrative results, highlighting the versatile competencies of our framework across diverse scenarios, are depicted in Figure 2.

To summarize, our contributions are the following:
**(i)** We present a simple, yet effective, scalable cross-modal framework to empower LLMs to adeptly handle a diverse range of tasks across a variety of modalities. Our results show that while each of the modalities (images, video, audio, and 3D) undergoes individual alignment to LLMs, our instruction-aware representations are generalizable for combined modality tasks, facilitating the seamless integration of more modalities as they emerge.
**(ii)** We introduce an approach for crafting instruction-tuning datasets from a variety of modalities, leveraging only readily available captioning data and open-source language models. Illustratively, we transform the established Cap3D (Luo et al., 2023) and AudioCaps (Kim et al., 2019) datasets into expansive QA collections, resulting in 250,070 and 31,020 entries, respectively.
**(iii)** We construct `DisCRn`, the first instruction-based cross-modal discriminative crossmodal reasoning dataset for evaluation. Comprising 42k examples that span video, audio, 3D, and image modalities, this dataset challenges models to distinguish between input modalities, thus establishing a new benchmark in this intricate domain.

## 2 RELATED WORK

**Multimodal Language Models (MLMs):** Recent years have seen a surge in the exploration of models capable of executing a spectrum of vision-language tasks, leading to the development of Multimodal Language Models (MLMs). These models align multimodal inputs and the LLM's latent space through various techniques, such as autoregressive unified pretraining of transformer-based architectures with multimodal inputs (Cho et al., 2021; Yang et al., 2022; Wang et al., 2022; Li et al., 2022a) or vision-to-language projection learning (Dai et al., 2023; Alayrac et al., 2022; Chen et al., 2022a; Mañas et al., 2023; Tiong et al., 2023; Liu et al., 2023; Huang et al., 2023; Gong et al., 2023; Li et al., 2023b; Koh et al., 2023; Driess et al., 2023b; Wu et al., 2023). Such approaches have been expanded beyond images, for video (Zhang et al., 2023b; Li et al., 2023c), audio (Zhang et al., 2023a), and 3D (Xu et al., 2023b). This work extends the multimodal instruction tuning framework from (Dai et al., 2023) to encompass audio, 3D, and video modalities.

This research aligns with contemporary advancements focused on aligning multiple non-linguistic modalities to LLMs, a concept we define in this paper as *Cross-Modal Models*. Strategies in this domain have often emulated those deployed in MLMs, involving the conceptualization of unified pretraining techniques (Wang et al., 2023; Chen et al., 2023b; Xu et al., 2023a; Shukor et al., 2023) and, more pertinently to our work, the alignment of various modalities with pretrained LLMs. The alignment to pretrained LLMs offers an economical advantage, optimizing the use of data and computational resources. Among the prevailing methods for modality alignment to pretrained LLMs is to leverage of a shared embedding space across different modalities, like ImageBind (Girdhar et al., 2023). Here, the LLM's projection is fine-tuned through a low-rank approach (Hu et al., 2021) primarily using one modality (most often vision due to its abundant datasets) and subsequently generalizing it to others (Su et al., 2023; Han et al., 2023). In a vein more directly aligned with our work, ChatBridge (Zhao et al., 2023) trains distinct projections for each modality while preserving the LLM weights frozen. Although each projection's preliminary alignment is executed independently using noisy data, it subsequently undergoes instruction tuning on artificially crafted joint modality data. This process can be resource-intensive, especially when leveraging API-based LLMs such as GPT-4 (OpenAI, 2023). Similarly to ChatBridge, we preserve the merits of a frozen LLM. However, we employ singular modality-to-text datasets for the individualized tuning of each projection and harness open-source LLMs to craft additional instruction-tuned data in a manner that is both replicable and economically efficient.

**Multimodal Multi-Input Language Tasks:** The advancements in single input vision-language tasks have paved the way for the development of tasks necessitating models to concurrently reason about multiple non-linguistic inputs. These tasks may require models to engage in spatial reasoning across multiple images (Bansal et al., 2020), deliberate over a series of slides (Tanaka et al., 2023), respond to queries necessitating cross-modal reasoning across images and tables (Li et al., 2022c), or execute a range of instruction-based tasks involving multiple image inputs (Li et al., 2022c). Despite their complexity, these tasks predominantly operate within the realms of image-text modalities.

Some tasks necessitate reasoning over videos (Chen et al., 2023a; Alamri et al., 2018; Li et al., 2022b), demanding models to congruently process both video and audio modalities. However, to our best knowledge, there exists no task that requires contrastive reasoning across disparate modalities in an open vocabulary text generation setting, such as 3D with images, or audio with video. Addressing this gap, we introduce `DisCRn` the first dataset of its kind. This dataset mandates models to comprehend that the instructions entail a comparative analysis of the diverse modality inputs and to subsequently generate a corresponding answer.

## 3 MODEL ARCHITECTURE

Figure 1 depicts an overview of the model's architecture which extends the instruction-aware projection method introduced in (Dai et al., 2023) to an arbitrary number of modalities. For any given modality $M$, we employ a pre-trained encoder $\text{Enc}_M$ to embed the raw input from the modality's input space $\mathcal{X}_M$ into an embedding space $\mathcal{Z}_M$. Formally, this encoding process can be represented as $z_M = \text{Enc}_M(x_M)$, where $x_M \in \mathcal{X}_M$ is an instance from the raw input of modality $M$. The encoder's output $z_M \in \mathcal{Z}_M$ is subsequently fed into the Q-Former module, specifically to its cross-attention layers. This Q-Former module also receives a set of $K$ learnable query embeddings $\boldsymbol{Q}_M = \{\boldsymbol{q}_{M_1} \dots \boldsymbol{q}_{M_K}\}$ and the task instruction $i_M \in \mathbb{I}_{M_t}$, where $\mathbb{I}_{M_t}$ is the predefined space of instruction templates for task $t$ and modality $M$. For a full list of templates refer to Appendix H.

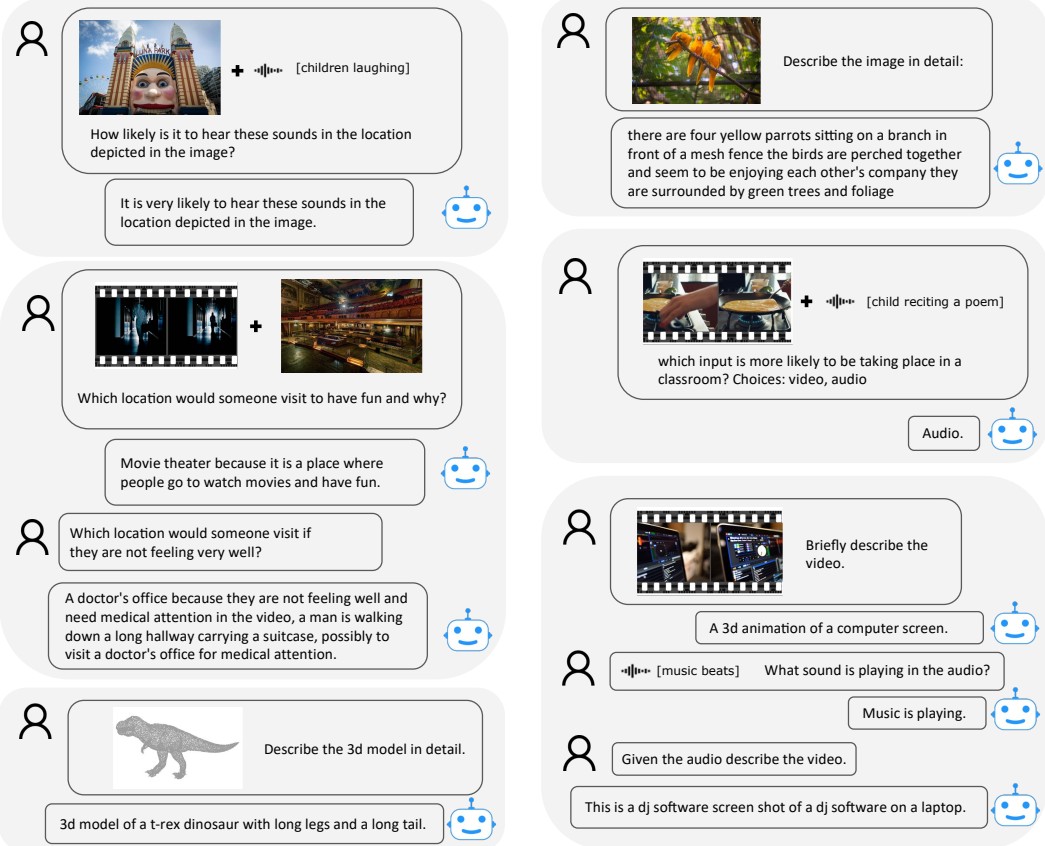

Figure 2: Qualitative Examples: `X-InstructBLIP` demonstrates multifaceted capabilities across the four examined modalities: image, video, audio, and 3D. It proficiently navigates tasks from image and 3D captioning to sophisticated cross-modal reasoning. This encompasses both discriminative and joint reasoning scenarios, showcasing its versatile applicability and nuanced understanding in diverse contexts.

Each Q-Former transforms the set of $K$ query embeddings, $\boldsymbol{Q}_M$, conditioned on both $(z_M, i_M)$, into instruction-aware language representations of the input modality. The output query tokens $\boldsymbol{Q}_M$ are linearly projected to the frozen LLM's space through a learnable projection layer $\mathrm{LP}_M$ specific to each modality. This step is necessary since the Q-Former's tokenization space differs from that of the frozen pretrained LLM. For sequential data, such as video and audio, we extract $N \times K$ query tokens; each frame is encoded and then processed separately by the Q-Former module. The Q-Former module consists of two transformer submodules that share the same self-attention layers: one submodule interacts with the output of the modality encoder $\mathcal{Z}_M$ and the other is a BERT$_{\text{base}}$ text transformer that serves as both an encoder and decoder. We initialize each Q-Former with the pre-trained weights from BLIP2 (Li et al., 2023b), without the cross-attention layers due to a dimension mismatch between our encoder modalities and the image encoder in BLIP2. The modality embedding $z_M$ interacts with the instruction text $i_M$ and learnable query tokens $\boldsymbol{Q}_M$ via cross-attention layers inserted in every other transformer block.

Finally, the output query tokens $\boldsymbol{Q}_M$ are linearly transformed to the LLM's space $\mathrm{LP}_M(\boldsymbol{Q}_M)$. Let TOK be the tokenizer associated with the LLM, EMB the LLM's embedding layer, $c_M$ the modality cue, $x$ the example text input, and $y$ the corresponding target phrase. The resulting input to the frozen LLM is formed by $\mathrm{EMB}(\mathrm{TOK}(c_M)) \oplus \mathrm{LP}_M(\boldsymbol{Q}_M) \oplus \mathrm{EMB}(\mathrm{TOK}(i_M)) \oplus \mathrm{EMB}(\mathrm{TOK}(x))$ where $\oplus$ denotes concatenation. The Q-Former is trained by minimizing the LLM cross-entropy loss on the output tokens, where the target is $\mathrm{TOK}(y)$.

## 4 DATASETS

We now discuss the pre-existing data suite and automatically generated datasets for instruction tuning, including our data augmentation technique to generate instruction tuning data for modalities with scarce data (Section 4.1). Furthermore, we introduce the Discriminatory Cross-modal Reason-

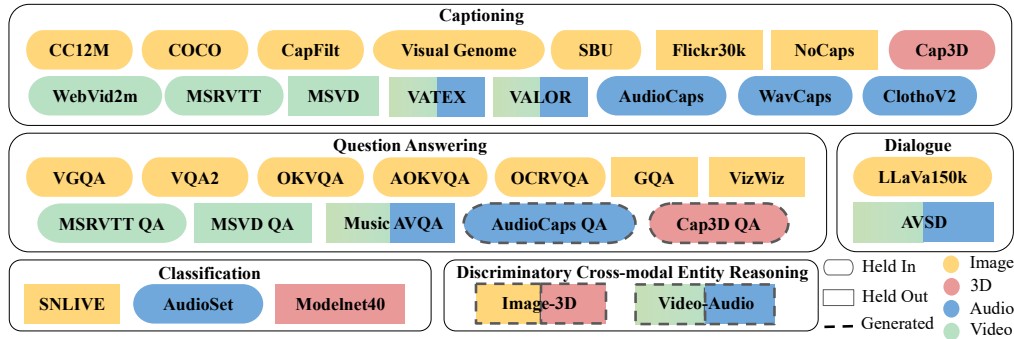

Figure 3: Instruction Tuning Datasets: This is the curated collection of publicly accessible datasets used for X-InstructBLIP's training and evaluation. Oval-enclosed datasets are training datasets, while square cells signify zero-shot evaluation sets. Datasets with a dashed outline are derived automatically from existing public datasets, using our multi-round prompting technique.

ing challenge dataset DisCRn used to evaluate the emergent capabilities of X-InstructBLIP (S [2] (Section 4.2).

## 4.1 FINETUNING DATASETS

**Existing Datasets:** Figure 3 illustrates the datasets utilized for both instruction finetuning and evaluation. A detailed breakdown of the dataset statistics and formats can be found in Appendix G. For each dataset in $\mathbb{D}_M$, the collection of held-in datasets specific to modality $M$, we adopted the sampling methodology described in (Dai et al., 2023), but extended to accommodate a broader range of modalities. The sampling probability for any given dataset $D_{M_d} \in \mathbb{D}_M$ is given by the equation $\frac{\sqrt{|D_{M_d}|}}{\sum_{d \in [1...|\mathbb{D}_M|]} \sqrt{|D_{M_d}|}}$. In the final $40,000$ iterations of the image Q-Former's training, we adjusted the sampling rate of MSCOCO to $3.0$, while retaining the ratios for other datasets. We set the sampling rate of MSRVTT to $1.0$ for the video Q-Former while maintaining the other sampling rates consistent with the formula. These adjustments guide the model to learn from a more refined but diverse dataset.

**Instruction Data Augmentation:** Extracting instruction-aware representations necessitates diverse instruction-related tasks across all modalities. Notably, datasets for 3D and audio modalities are largely caption-centric. To address this, we leverage the open-source large language model google/flan-t5-xxl (Chung et al., 2022) from the huggingface-transformers platform to automatically generate question-answer pairs for the 3D and audio modalities, based on their corresponding captions. The process begins by prompting the model with captions to generate potential answers. These answers are then used to prompt the model to generate candidate questions. If the model's response to a question, using the caption as context, aligns closely with the initial answer (achieving a Levenshtein similarity score (Yujian & Bo, 2007) above 0.9), the example is added to our dataset. This procedure yields 250,070 examples using 3D data from Cap3D (Luo et al., 2023) [3] and 31,020 examples for audio data from AudioCaps (Kim et al., 2019). See Appendix A.1 for a detailed breakdown of the data generation process and the resulting distribution.

## 4.2 DISCRN: DISCRIMINATIVE CROSS-MODAL REASONING DATASET

X-InstructBLIP offers a distinct emergent capability: reasoning across different modalities, even when each modality Q-Former is trained individually. This highlights the model's versatility and potential scalability across numerous modalities. To underscore this cross-modal reasoning capability, we present a pioneering cross-modal discriminatory reasoning test dataset. As illustrated in Figure 4, this dataset challenges the model to discern between the properties of two entities across modalities by selecting which one satisfies a queried property. Undertaking this task mandates the

---

[2] *Discriminative reasoning* is the ability to distinctly discern the relationships between pairs of inputs provided to the model. We adapt the definition from Xu et al. (2021), where it is applied it for single-modality text pair paragraphs. In our context, the model is required to reason across disparate modalities, such as video-audio and image-3D. We distinguish discriminative reasoning from *joint reasoning* which, in the context of this work, denotes the ability of models to synthesize information from aligned cross-modal inputs to accomplish a task.

[3] A subset consisting of 5k point clouds is deliberately held-out from Cap3D for the construction of DisCRn, as introduced in Section 4.2. This exclusion is maintained both in the captioning and QA configurations.

Figure 4: Sample examples from `DisCRn`. Given two distinct modality inputs, the model needs to select the entity that matches the property queried. Audio is symbolized by waveforms and its semantics are conveyed via annotated captions, 3D is illustrated through a point cloud visualization, and videos are represented through the display of two random frames.

model to not only discriminate the inherent characteristics of the involved modalities but also to consider their relative positioning in the input. This strategic imposition serves to diminish reliance on simplistic text-matching heuristics or potential deceptive correlations between modalities, which could allow basic captioning baselines to circumvent the task.

To generate the dataset, we employed the `google/flan-t5-xxl` model, previously utilized for instruction data creation (refer to section 4.1). The process is initiated by prompting the language model in a Chain-of-Thought (Wei et al., 2022) manner to generate a set of properties for each dataset instance. Each instance is then paired with a random entity from the dataset to form a (question, answer, explanation) triplet by prompting the language model with three in-context examples. A pivotal step in this creation process is a round-trip-consistency check: an example is only integrated into the final dataset when the model's predictions on the generated question, given the captions, match the example answer, exhibiting a Levenshtein distance above 0.9. This refined dataset encompasses 12,162 audio-video samples sourced from the AudioCaps validation set, and 29,072 image-point cloud instances derived from a reserved subset of 5k point clouds from Cap3D (Luo et al., 2023). Each instance in the dataset is coupled with two representations corresponding to the captions—audio-video from AudioCaps and point cloud-images from Cap3D. Given that the arrangement of the data can be altered, this allows for maintaining a balanced set of answers. This balance pertains not only to the position of the answers but also to the modality, Human raters find the dataset exhibits a commendable quality, achieving a 90% accuracy rate across the different modalities. See Appendix A.2 for a thorough discussion of the dataset's creation and distribution.

## 5 EXPERIMENTS

Section 5.1 provides the implementation details for our experiments, followed by the results for generation and classification tasks involving an individual modality alongside the text in section 5.2. Section 5.3 explores `X-InstructBLIP`'s emergent ability to competently handle tasks involving cross modality reasoning, even with modalities trained independently.

### 5.1 IMPLEMENTATION DETAILS

Our model is built on the LAVIS library's framework (Li et al., 2023a). Each Q-Former possesses 188M trainable parameters, a hidden dimension of size 768, and $K = 32$ query tokens. For image and video modalities, the chosen pre-trained encoder is `eva_clip_g` model (Fang et al., 2022). For audio, we utilize `beats` (Chen et al., 2022b), and for 3D we opt for `ulip2` with a `pointBERT` backbone (Xue et al., 2023). Standardized preprocessing is applied for each of the modalities. Images are resized to $224 \times 224$ pixels. Audio files undergo mono conversion and features extraction from 5-second frames, resulting in 2 frames. Video frames are processed similarly to images, with 5 and 2 frames sampled uniformly per video for individual and crossmodal tasks respectively. For 3D point clouds, 8,192 points are uniformly sampled as in ULIP-2 Xue et al. (2023). We maintain consistent generation templates for the evaluated tasks, with small modifications tailored to the particular modalities. Table 14 (Appendix) has details on evaluation hyperparameters and prompts. We report our experimental results atop of the Vicuna 7b and 13b models (Chiang et al., 2023).

In terms of optimization, we use AdamW (Loshchilov & Hutter, 2017) with parameters $\beta_1 = 0.9$ and $\beta_2 = 0.999$, and a weight decay of 0.05. The learning rate warms up linearly over the initial 1,000 steps from $10^{-8}$ to $10^{-5}$, followed by a cosine decay to a minimum of 0. The computational experiments are performed on configurations with 8 A100 40GB GPUs. We select a single best model for each modality Q-Former as delineated in Appendix E.

## 5.2 RESULTS: INDIVIDUAL MODALITY UNDERSTANDING

We evaluate `X-InstructBLIP`'s performance across a range of single modality to text tasks, illustrating its versatility and efficacy across all four explored modalities. Tables 1, 2, and 3 summarize the `X-InstructBLIP`'s performance across image, audio and video, and 3D modalities.

**Image modality:** `X-InstructBLIP` attains SoTA results in zero-shot VizWiz (Bigham et al., 2010) while performing comparably to InstructBLIP across all tasks evaluated. We note a small drop in performance compared to InstructBLIP despite the similar finetuning setup. We hypothesize that this minor decrement in performance is attributed to the expanded template space which introduces a trade-off of generalization and performance since InstructBLIP's ability to generate more precise representations from a compact set of templates but fails to handle a larger variation of templates that were not seen in training. In table 16 (Appendix) we show that while `X-InstructBLIP` is more robust to template variations than InstructBLIP, it still endures a noticeable drop in performance.

**Video modality:** `X-InstructBLIP` establishes new benchmarks for zero-shot performance in MSVD Captioning (Chen & Dolan, 2011) and MSVD QA (Xu et al., 2017). Intriguingly, video is the sole modality where the incorporation of modality *cues* during both training and inference doesn't invariably enhance performance. This discrepancy may stem from the relatively extensive number of frames in video compared to other modalities, coupled with recognized biases in video-language tasks (Buch et al., 2022). Such conditions might generate a spurious correlation, prompting the model to primarily rely on initial frames for output -- a strategy potentially advantageous for certain tasks but detrimental for others. A more in-depth exploration of this occurrence is reserved for subsequent research, as the current paper's focus is to demonstrate the universal efficacy of our framework across modalities. The improvement over InstructBLIP and a more detailed ablation study in Appendix C shows that there is performance benefit in training the Q-Former explicitly on videos instead of relying only on the image alignment.

**Audio modality:** For audio tasks, it attains SoTA zero-shot performance on ClothoV2 (Drossos et al., 2021). Remarkably, `X-InstructBLIP` surpasses ChatBridge in both audio and video zero-shot domains. This is noteworthy as ChatBridge also utilizes Q-Formers to learn modality projections to LLMs but lacks instruction awareness, underscoring the crucial role of instruction-aware representations in achieving robust modality-to-language alignments.

**3D Modality:** Our model displays comparable performance to its 3D encoder backbone in closed vocabulary classification settings applied using the loss ranking method described in (Li et al., 2021). It also sets new standards in zero-shot performance for open generation settings, validated by its accuracy in identifying the correct ModelNet40 (Chang et al., 2015) class within object descriptions when prompted with `Describe the 3D model`. Our model demonstrates substantial improvements over the InstructBLIP baseline, which processes a single view rendering of the point cloud. Notably, our approach surpasses the PointLLM (Xu et al., 2023b)—which is constructed on the Vicuna13b model and includes an RGB encoder—by a notable margin of 4.8/5.2 points with `X-InstructBLIP` (7b) and (13b) respectively.

## 5.3 EMERGENT ABILITY: CROSS-MODAL UNDERSTANDING

### 5.3.1 VIDEO-AUDIO JOINT REASONING

Despite each modality projection being trained individually, `X-InstructBLIP` shows strong joint and cross-modal reasoning abilities. Table 4 demonstrates `X-InstructBLIP`'s capability to reason jointly over video (V) and audio (A), showcasing comparable performance with models finetuned with integrated video-audio data. Notably, `X-InstructBLIP` (7b) excels in synergizing inputs, displaying an improvement in performance compared to utilizing a single modality, a phenomenon less accentuated in `X-InstructBLIP` (7b) *'w/o cue'*. Finally, `X-InstructBLIP` performs comparatively to ChatBridge (Zhao et al., 2023), a Vicuna13b based cross-modal model finetuned on joint video-audio data. In the Music AVQA (Li et al., 2022b) `X-InstructBLIP`

---

[4] Models explicitly finetuned on COCO and report out-of-distribution captioning performance on NoCaps. In contrast, `X-InstructBLIP` only partly samples COCO during the Q-Former alignment tuning.

[5] The best value among all InstructBLIP variations is reported with the exception of SNLI-VE where only the Vicuna-based InstructBLIP variations are considered, since FlanT5 models (Chung et al., 2022) have been finetuned on a text-only version of SNLI-VE(Camburu et al., 2018).

| | Image | Audio | Video | 3D | Image Captioning | | | Visual Question Answering | | Visual Entailment |
|---|---|---|---|---|---|---|---|---|---|---|
| | | | | | Flickr30k | NoCaps | | VizWiz | GQA | SNLI-VE |
| | | | | | test | out-domain | val all | test-dev | balanced test-dev | test |
| Flamingo 9B(Alayrac et al., 2022) | ✓ | ✗ | ✓ | ✗ | 61.5 | - | - | 28.8 | - | - |
| Flamingo 80B | ✓ | ✗ | ✓ | ✗ | 67.2 | - | - | 31.6 | - | - |
| BLIP-2(T5xl)(Li et al., 2023b) | ✓ | ✗ | ✓ | ✗ | 76.1 | 111.7 | 104.5 | 29.8 | 44.2 | - |
| BLIP-2(T5xxl) | ✓ | ✗ | ✓ | ✗ | 73.7 | 113.1 | 98.4 | 29.4 | 44.7 | - |
| BLIP-2(7b) | ✓ | ✗ | ✓ | ✗ | 74.9 | - | 107.5 | 25.3 | 38.6 | - |
| BLIP-2(13b) | ✓ | ✗ | ✓ | ✗ | 71.6 | - | 103.9 | 19.6 | 41.0 | - |
| mPLUG(Li et al., 2022a) | ✓ | ✗ | ✓ | ✗ | - | 117.8[4] | 114.8[4] | - | - | 89.3 |
| KOSMOS-1(Huang et al., 2023) | ✓ | ✗ | ✗ | ✗ | 67.1 | - | - | 29.2 | - | - |
| InstructBLIP[5](Dai et al., 2023) | ✓ | ✗ | ✓ | ✗ | **84.5** | **122.5** | **123.1** | 34.5 | **49.5** | 58.7 |
| UnifiedIO$_{XL}$(Lu et al., 2022) | ✓ | ✓ | ✓ | ✗ | - | - | 100.0 | 57.4 | - | - |
| UniVAL(Shukor et al., 2023) | ✓ | ✓ | ✓ | ✗ | - | 95.3[4] | - | 20.2 | 78.2 | - |
| ChatBridge(13b)(Zhao et al., 2023) | ✓ | ✓ | ✓ | ✗ | 82.5 | - | 115.7 | - | 41.8 | - |
| PandaGPT(7b)(Su et al., 2023) | ✓ | ✓ | ✓ | ✓ | - | - | - | 29.7 | 41.6 | |
| ImageBindLLM(13b)(Girdhar et al., 2023) | ✓ | ✓ | ✓ | ✓ | - | - | 30.4 | - | 41.2 | |
| LLaVA(7b)(?) | ✓ | ✓ | ✓ | ✓ | 27.7 | - | 33.1 | - | 41.3 | 57.8 |
| X-InstructBLIP (7b) | ✓ | ✓ | ✓ | ✓ | 82.1 | 115.5 | 117.5 | **34.9** | 48.1 | 57.6 |
| X-InstructBLIP (7b) w/o cue | ✓ | ✓ | ✓ | ✓ | 82.2(↑0.1) | 113.7(↓1.8) | 115.6(↓1.0) | 33.9(↓1.6) | 46.6(↓1.5) | 57.0(↓0.6) |
| X-InstructBLIP (13b) | ✓ | ✓ | ✓ | ✓ | 74.7 | 114.5 | 36.0 | 34.8 | 49.2 | **58.9** |

Table 1: Image Zero-Shot Quantitative Results: CIDEr score is reported for captioning tasks and accuracy for all other tasks. Gray shaded rows correspond to current work. Gray rows represent models from this study. Underlined, **Bold**, and Blue numbers denote fine-tuned evaluations, top, and second-best zero-shot performances, respectively. (7b) and (13b) indicate the underlying Vicuna model size. *'w/o cue'* indicates that the model was trained and evaluated without specifying the type of modality provided in the query output tokens. This notation is followed in all subsequent tables.

| | MSVD | Video VATEX | MSVD QA | Audio Clotho | |
|---|---|---|---|---|---|
| | test | val | test | eval (v1) | val (v2) |
| VALOR(Chen et al., 2023a) | 15.6 | 95.8 | 60.0 | 8.0 | - |
| VAST(Chen et al., 2023b) | - | 99.5 | - | 50.7 | 51.9 |
| ChatBridge(13b)(Zhao et al., 2023) | - | - | 45.3 | - | 26.2 |
| UniVAL(Shukor et al., 2023) | - | - | 27.5 | - | 38.0 |
| mPLUG-2(Xu et al., 2023a) | **148.2** | - | 55.3 | - | - |
| InstructBLIPDai et al. (2023)[5] | 87.2 | 57.6 | 41.2 | - | - |
| X-InstructBLIP (7b) | 116.1 | **59.2** | 51.7 | **29.3** | 27.4 |
| X-InstructBLIP (7b) w/o cue | 118.7 | 59.5 | 50.5 | 26.9 | 24.5 |
| X-InstructBLIP (13b) | **113.3** | 52.0 | 49.6 | 28.7 | **27.4** |

Table 2: Zero-Shot quantitative results for audio or video individual modality to language tasks.

outperforms ChatBridge by using Vicuna 7b and Vicuna13b respectively, while it lags behind in AVSD (Alamri et al., 2018) and VALOR (Chen et al., 2023a) respectively.

### 5.3.2 DISCRN EVALUATION

We assess the ability of X-InstructBLIP in executing discriminatory reasoning across different modalities using our newly introduced DisCRn benchmark, detailed in Section 4.2. We frame this problem as a realistic open vocabulary generation task, allowing the model to independently formulate the output but steering the generation via a structured template: ``question {question}? options: first, second answer:''. While the language model instruction includes the answer options, these are not supplied as inputs to the modality Q-Formers. This strategy aims to foster representations that are more attuned to the posed question, avoiding spurious associations with positional information unavailable to individually trained Q-Formers.

| | Modelnet40 (test) | | | |
|---|---|---|---|---|
| Classification | | Open Vocabulary Generation | | |
| ULIP2 (Xue et al., 2023) | **66.7** | Point LLM (+ RGB) (Xu et al., 2023b) (13b)† | | 44.8 |
| InstructBLIP (7b) [Single View] (Dai et al., 2023) | 31.4 | InstructBLIP (7b) [Single View ] † | | 23.7 |
| InstructBLIP (13b) [Single View] (Dai et al., 2023) | 31.5 | InstructBLIP (13b) [Single View] † | | 25.5 |
| X-InstructBLIP (7b) | 60.9 | X-InstructBLIP (7b)† | | 49.6 |
| X-InstructBLIP (7b) w/o cue | 62.8 | X-InstructBLIP (7b) w/o cue† | | 49.4 |
| X-InstructBLIP (13b) | 65.1 | X-InstructBLIP (13b)† | | **50.0** |

Table 3: 3D Zero-Shot Quantitative Results: † indicates open-vocabulary generation, as opposed to loss ranking classification (Li et al., 2021).

| Model | A+V FT | AVSD dstc7-test | | | Music AVQA test | | | VALOR test | | |
|---|---|---|---|---|---|---|---|---|---|---|
| | | A+V | V | A | A+V | V | A | A+V | V | A |
| ChatBridge(13b) | ✓ | **75.4** | **73.1** | 46.2 | 43.0 | 33.1 | 28.9 | **24.7** | **22.3** | 5.2 |
| `X-InstructBLIP` (7b) | ✗ | 46.8 | 41.5 | 34.2 | 28.1 | 27.2 | 13.4 | 20.1 | 19.7 | 7.5 |
| `X-InstructBLIP` (7b) *w/o cue* | ✗ | 43.5(↓3.3) | 35.8(↓5.7) | 26.5(↓9.9) | 22.3(↓6.8) | 27.3(↑.1) | 8.9(↓4.5) | 18.2(↓1.9) | 19.2(↓0.5) | **7.6**(↑0.1) |
| `X-InstructBLIP` (13b) | ✗ | 66.3 | 54.0 | **48.5** | **44.5** | **43.5** | 22.7 | 16.1 | 18.0 | 6.9 |

Table 4: Emergent joint video(V)-audio(A) reasoning. Despite individual modality training, `X-InstructBLIP` achieves comparable performance with models trained on joint video-audio data.

| `DisCRn` | Image-3D | Audio-Video |
|---|---|---|
| Caption Baseline (7b) | 41.8 | 30.8 |
| Linear Projection$_{3D/Audio}$ | 19.7 | 22.3 |
| X-InstructBLIP (7b)† | 57.7 | 31.4 |
| X-InstructBLIP (7b) | 48.1 | 34.0 |

Table 5: `DisCRN` evaluation.

We conduct additional experiments where we provide a distinct instruction to the Q-Former from that provided to the LLM. In particular, we condition the each modality Q-Former to the instruction ``Describe the {modality}.'' denoted by `X-InstructBLIP`*. to prime the Q-Former to output more descriptive query tokens. This is crucial since Q-Formers, not being trained on cross-modal tasks, may not formulate representations that are adequately informative to address the task.

To benchmark our model's capabilities, we incorporate a robust captioning baseline and prompt the Vicuna model with captions corresponding to the modalities, substituting the query outputs from `X-InstructBLIP`. For images, 3D, and video modalities, we elicit captions from Instruct-BLIP (Dai et al., 2023). Specifically, for 3D inputs, a randomly rendered grayscale view of the model serves as input to InstructBLIP, aligning with our 3D encoder ULIP2's (Xue et al., 2023) incapability to interpret color. Captions for videos are derived from five uniformly sampled frames, and for audio, the WavCaps model (Mei et al., 2023) is employed. Since it is an open vocabulary generation setup,the random output is $\frac{1}{|\mathbb{V}|}^{|l|}$, where $l = 10$ is the maximum output length allowed by the model and $|\mathbb{V}| =$32k is the vocabulary size[6]. The model employs Beam Search with a beam size of 5 and a length penalty of -1, facilitating the generation of concise responses.

Table 5 provides a summary of the results from the discriminatory reasoning experiments. It is evident that `X-InstructBLIP` surpasses the captioning baseline in both Image-3D and Audio-Video categories, despite the inherent challenges of the task both in terms of crossmodality and language based positional reasoning. A noteworthy observation is the doubling of performance on the Image-3D subset when description-based query outputs from the Q-Former are introduced, contrasted by a decline in the performance of the audio-video category. This discrepancy is likely attributed to the extensive training duration of the image Q-Former due to the large amount of data, allowing it to fine-tune its responsiveness to instructions, while the sequential Q-Formers reach convergence swiftly, subsequently experiencing a dip in performance and thus preserving more similar representation content across different instructions.

## 6 CONCLUSION

This study presents a scalable framework for independently aligning multiple modalities to a static Large Language Model (LLM), achieving competitive performance, particularly in zero-shot settings, and establishing new State-of-the-Art benchmarks. The framework exhibits emergent cross-modal reasoning, evaluated through the introduced `DisCRn` dataset, emphasizing the universal effectiveness of instruction-aware representations across varied modalities. However, it also reveals complexities and unanswered questions within each modality, paving the way for future exploration into the formation of instruction-aware representations in distinct modalities.

---

[6] For simplicity, we assume a random probablity distribution of the output tokens, ignoring the model distribution to distinguish from a potential binary classification set up.

## 7 REPRODUCIBILITY STATEMENT

In alignment with the principles of open science and to foster reproducibility, transparency, and further research, we promise to provide open source access to all the resources associated with our study, including: a complete, documented, and public codebase with all the scripts, models, preprocessing, and evaluation code necessary to replicate the experiments. We will be further releasing the pretrained model weights along side the exact evaluation configs that generated the results cited in the paper. We show our commitment to reproducibility through an extensive Appendix that highlights details of training and evaluation. Furthermore, all experiments were completed with prespecified random seeds that will also be made available in the experiment configuration files. Finally, we will release all datasets collected for this study for public download, as well as the code used to generate them. In addition to providing these resources, we pledge to maintain them and offer requisite support for any queries or clarifications related to the provided resources, contributing to a supportive and inclusive research environment.

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

# A  DATA GENERATION

## A.1  INSTRUCTION TUNING DATA AUGMENTATION

For the audio and 3D modalities, the available range of tasks for instruction tuning is relatively limited. To address this challenge we follow a common paradigm in the literature (Xu et al., 2017) and extract question-answer pairs are derived from captioning datasets, specifically from captions consisting of 10 words or more. Figure 5 delineates the procedure to automatically generate question answering data from captioning datasets. The `google/flan-t5-xxl` model from huggingface-transformers is employed, and is prompted to produce candidate single-word answers based on the caption. Subsequently, the model is tasked with generating a relevant question using the answer and context as inputs. The method of round-trip-consistency (Paranjape et al., 2021) is utilized to sift through and retain only those question-answer pairs that align with the context. This alignment is verified by ensuring that the Levenshtein partial similarity between the predicted and initial answers is greater than 0.90, calculated using the Fuzzy Wuzzy Python package. Subsequently, we apply a string matching post-processing to filter out instances that do not conform to the prescribed format. As a result, 250,070/1,157 suitable training/validation examples are derived from an initial 661,576/5,000 3D-caption samples from the Cap3D (Luo et al., 2023) dataset, and 31,020/2,475 training/validation examples are derived from 49,838/2,475 original audio-caption samples from the AudioCaps (Kim et al., 2019) dataset. Moreover, for 3D data, it is imperative to ensure that the question-answer pairs do not allude to color. This is due to the fact that the 3D encoder does not capture color characteristics. To achieve this, the language model is directed to reformulate the captions by omitting any references to color, prompted as: `Rewrite the sentence {caption} by eliminating any color mentions`, prior to implementing the round-trip-consistency check. A short human evaluation on 50 samples for each modality shows that 90% of the generated audio and 82% of the 3D data is correct. Table 6 presents a random sample of the generated data and table 7 provides an overview of the datasets's distribution statistics. It is worth noting that the error cases are typically due to non-sensical questions, rather than wrong answers. For example the following pairs were marked as non-sensical: *What is the sewing machine running at? speed, What does the steam whistle do? hisses, What is the 3D model of a brick wall with holes and stacked cubes, resembling? elements*, and *What is the hat with? pattern*.

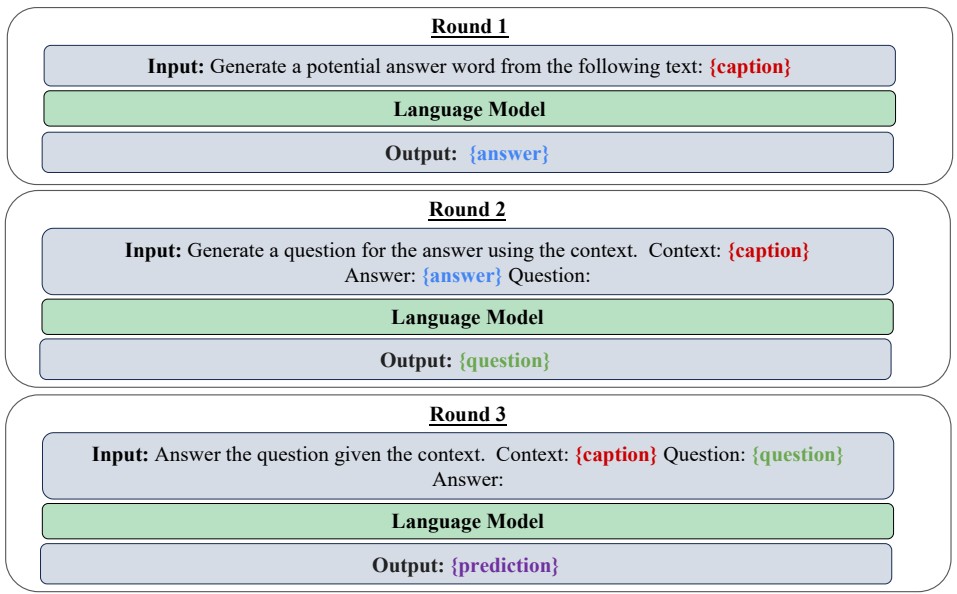

Figure 5: Round-Trip-Consistency Prompting for QA Datasets in 3D and Audio Modalities.

| Modality | Caption | Question | Answer |
|---|---|---|---|
| Audio | A woman speaks while types a keyboard; | What is the woman typing on? | Keyboard |
| | A man are talking while multiple dogs are barking around them; | What is the dog doing? | Barking |
| | A man speaks and a crowd applauds, he continues talking; | What does the crowd do after the man speaks? | Applauds |
| | A plane flies in the distance as a man speaks and metal clinks. | What does the metal do? | Clinks |
| 3D | A 3D model of a wooden chair and stool with a chained bucket on it | What is on the stool? | Bucket |
| | A 3D model of a moss-covered stone, resembling a leaf, paper map, and rock | What is covering the stone? | Moss |
| | A balloon with a string attached, featuring a teddy bear and a cat face on it | What is the object with a string attached? | Balloon |
| | A 3D model of various food items,including an oyster, a piece of fruit, and different forms of eggs. | What is the food item that is a shellfish? | oyster |

Table 6: Automatically Generated QA examples from Captioning Data.

| Dataset | AudioCapsQA | | Cap3DQA | |
|---|---|---|---|---|
| | *train* | *val* | *train* | *val* |
| Total Size | 31,020 | 1,653 | 250,070 | 1,157 |
| Number of Distinct Questions | 12,174 | 1,003 | 67,001 | 953 |
| Number of Distinct Answers | 1,837 | 423 | 4,555 | 451 |
| Average Question Length | 6.0 words | 6.1 words | 6.8 words | 7.0 words |
| Question Vocabulary Size | 3,260 words | 808 words | 12,771 words | 1,022 |

Table 7: QA Generated Dataset Statistics

## A.2 CROSS-MODAL DISCRIMINATIVE REASONING DATA GENERATION

To assess the cross-modal reasoning capabilities of `X-InstructBLIP`, we devised a unique task that repurposes existing captioning datasets, specifically focusing on data representable in multiple modalities. We chose the AudioCaps (Kim et al., 2019) validation dataset and reserved a subset of 5k examples from Cap3D (Luo et al., 2023) as our validation dataset, ensuring that the 3D Q-Former is not exposed to this subset during the training phase in either captioning or 3DQA settings.

The audio data from AudioCaps originates from Youtube videos, allowing us to download the corresponding video files using their YouTube IDs. For Cap3D, we employed the associated point clouds and randomly selected one rendered image from the available eight view angles within the dataset.

A depiction of the data generation procedure, also outlined in the main text, is provided in Figure 6. During the evaluation, we maintain a balance, ensuring each option (A or B) serves as the ground truth 50% of the time. Given that this problem is structured as an open vocabulary generation task, we expanded the ground truth answer space to include synonyms and equivalent expressions, such as `[{answer modality}, left, 1st, 1, first, input 1, entity 1, object 1, input A, entity A, object A, A]` and `[{answer modality}, right, 2nd, second, input 2, entity 2, object 2, input B, entity B, object B, B]`, corresponding to whether the first or the second input is the ground truth. The human performance on a subsample of 100 examples of the dataset is 90%. Figure 4 in the main paper presents a sample of the generated data and table 8 provides an overview of the datasets's distribution statistics.

## B  Q-FORMER OUTPUT MODALITY CLUSTER FORMATION

We explore the effects on latent space distribution by fine-tuning the modality Q-Formers. Figure 7 shows a 2D visualization of the Q-Former query outputs for the same data points across two distinct modalities using the prompt "Describe the input.". Specifically, we initially apply a mean function

| Dataset | Audio-Video | Video-3D |
|---|---|---|
| Total Size | 12,160 | 28,173 |
| Number of Distinct Questions | 1,510 | 3,100 |
| Average Question Length | 7.1 words | 6.6 words |
| Question Vocabulary Size | 820 words | 1,272 words |

Table 8: `DisCRn`: Discriminative Cross-modal Reasoning Dataset Statistics

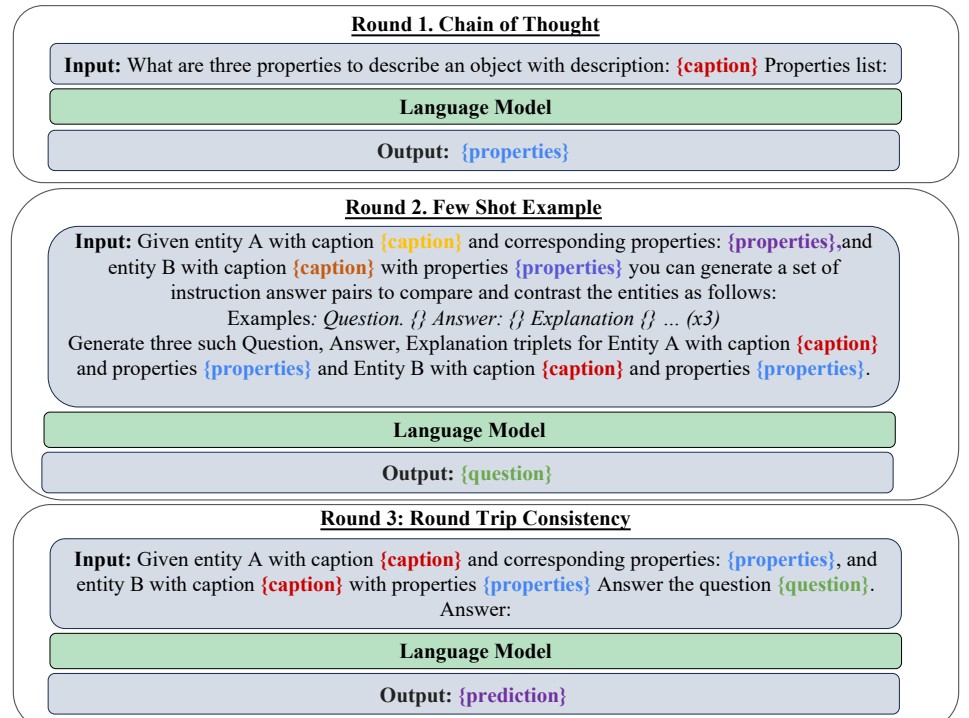

Figure 6: Cross-modal Discriminative Reasoning Dataset Generation Framework: The open source LLM is first prompted in a Chain-of-Thought manner to generate a set of properties matching the properties for the entire captioning dataset. For every instance within this dataset, another random entity is selected to generate a (question, answer, explanation) triple, by providing three in context examples. Finally, a round-trip-consistency step is performed such that only those instances whose predicted answers exhibit a Levenshtein distance exceeding 0.9 in comparison to the generated response are assimilated into the QA dataset.

along the language embedding dimension for each query output to summarize individual tokens and facilitate clustering. Subsequently, we employ T-SNE dimensionality reduction (Van der Maaten & Hinton, 2008) using the `scikit-learn` implementation.

For the Video-Audio modality, we select 10 classes from the validation set of VGGSound (Chen et al., 2020) and acquire the corresponding YouTube videos and audio clips. We sample 2 frames for both modalities to match their latent dimensions and facilitate visualization. In the case of Image-3D, we randomly sample 50 examples from each of 10 classes from ShapeNet (Chang et al., 2015), along with their associated rendered images.

The finetuned outputs exhibit a noticeable reorganization of data points within the latent space. Modalities form prominent clusters, and elements sharing identical labels tend to cluster together. Although some overlap persists, particularly within the audio modality, `X-InstructBLIP` appears to effectively associate data with shared semantic information. Additionally, the distinct training of Q-Formers and the utilization of diverse encoders seem to contribute to the distinct separation

| | Video-Audio | | 3D-Image | |
| --- | --- | --- | --- | --- |
| | Intra Distance↓ | Inter Distance↑ | Intra Distance↓ | Inter Distance↑ |
| Random Initialization | 1.1e-7 | 1.3e-8 | 7.4e-8 | 3.2e-9 |
| X-InstructBLIP | 5.2e-2 | 8.8e-3 | 2.2e-2 | 3.6e-3 |
| X-InstructBLIP *w/o cue* | 3.3e-2 | 6.6e-3 | 1.4e-2 | 3.6e-3 |

Table 9: Cluster Information

of modalities. Remarkably, the Image and 3D modalities, which share similar encoder architectures, exhibit closer clustering.

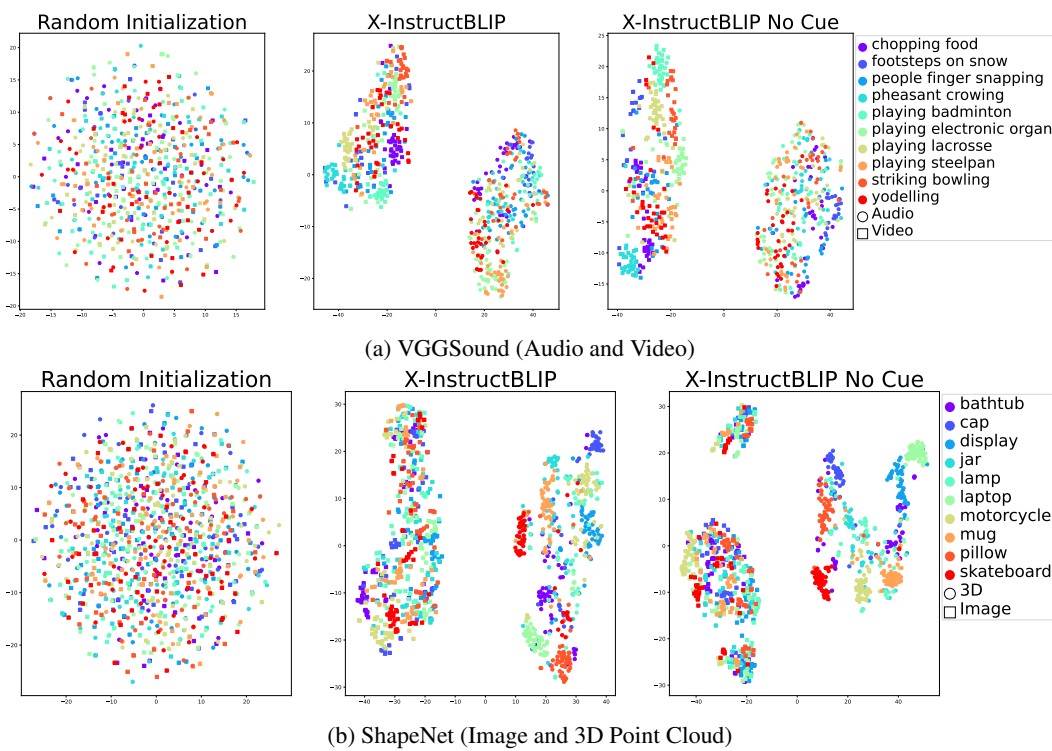

(a) VGGSound (Audio and Video)

(b) ShapeNet (Image and 3D Point Cloud)

Figure 7: T-SNE Visualization: Q-Former Query Outputs at different configurations.

## C  VIDEO Q-FORMER FINE-TUNING VERSUS IMAGE INITIALIZATION

To explore the impact of further training the Image Q-Formers on video data, Table 10 presents the results of evaluating video tasks using the weights from the Image Q-Formers. It is evident that training on video data enhances performance. However, it's worth noting that the Video Q-Formers reach convergence at an earlier stage (15k and 4k iterations for Vicuna7b and Vicuna13b, respectively). This is likely because the Q-Formers have already achieved semantic understanding during the image alignment phase, requiring minimal additional training to capture the nuances of sequential video projections. The higher drop in performance in MSVD (Chen & Dolan, 2011) captioning compared to VATEX (Wang et al., 2020) is likely due to the closer similarity between MSVD and MSRVTT (Xu et al., 2016) dataset distributions which is used for training.

## D  IN-DOMAIN EVALUATIONS

Table 11 presents in-domain performance for a sample of datasets seen in training across all four modalities. It's important to clarify that when we refer to 'in-domain,' we are specifically referring to datasets that were sampled during the training process. However, it's crucial to note that this does

| | MSVD | VATEX | MSVD QA |
|---|---|---|---|
| | *test* | *val* | *test* |
| `X-InstructBLIP` (7b) | 103.7 | **58.5** | 50.2 |
| `X-InstructBLIP` (7b) [*image*] | 37.6(↓66.1) | 28.4(↓30.1) | 40.6(↓9.6) |
| `X-InstructBLIP` (7b) *w/o cue* | 109.8 | 56.9 | 49.7 |
| `X-InstructBLIP` (7b) *w/o cue*[*image*] | 54.3(↓55.5) | 51.1(↓5.8) | 39.9(↓9.8) |
| `X-InstructBLIP` (13b) | 113.3 | 52.0 | 49.6 |
| `X-InstructBLIP` (13b) [*image*] | 69.7(↓63.6) | 52.0 | 36.7(↓12.9) |

Table 10: Impact of Training Image Q-Formers on Video. Models labeled as [*image*] utilize the Image Q-Former for video alignment.

not constitute explicit fine-tuning, as there is no guarantee that the Q-Former has encountered the entirety of the dataset during its training.

| | Image | | | 3D | | Video | | | | Audio | | |
|---|---|---|---|---|---|---|---|---|---|---|---|---|
| | OKVQA | COCO | | Cap3D | | MSRVTT | | MSRVTT QA | | AudioCaps | | |
| | *test* | *val* | *test* | *val* | *qa-val* | *val* | *test* | *val* | *test* | *val* | *test* | *qa-val* |
| Finetuned SOTA | 66.1 | | 155.1 | - | - | - | 80.3 | - | 49.6 | - | 80.6 | - |
| | (Driess et al., 2023a) | (Li et al., 2022a) | | | | | (Xu et al., 2023a) | (He et al., 2023) | | (Labbé et al., 2023) | | |
| InstructBLIP (T5xl) | 48.6 | 137.7 | 140.1 | - | - | 43.3 | 44.7 | 25.0 | 22.1 | - | - | - |
| InstructBLIP (T5xxl) | 47.7 | 138.9 | 140.7 | - | - | 40.6 | 41.3 | 25.6 | 22.3 | - | - | - |
| InstructBLIP (7b) | 57.2 | 141.2 | 142.0 | - | - | 28.2 | 30.7 | 22.1 | 18.8 | - | - | - |
| InstructBLIP (13b) | 56.3 | 139.1 | 141.1 | - | - | 38.1 | 37.8 | 24.8 | 20.2 | - | - | - |
| `X-InstructBLIP` (7b) | 52.5 | 137.1 | 137.9 | 145.6 | 57.8 | 59.4 | 55.5 | 42.4 | 42.1 | 71.5 | 68.0 | 40.9 |
| `X-InstructBLIP` (7b) *w/o cue* | 51.2 | 133.9 | 134.9 | 134.2 | 49.1 | 59.1 | 54.8 | 41.3 | 41.2 | 66.7 | 64.6 | 38.5 |
| `X-InstructBLIP` (13b) | 52.4 | 130.8 | 130.6 | 147.9 | 54.6 | 56.7 | 54.0 | 36.4 | 36.5 | 54.8 | 53.2 | 36.6 |

Table 11: In-Domain performance across modalities.

# E    TRAINING DETAILS

Table 12 compiles the training hyperparameters employed for each modality and model. The X-InstructBLIP –*prefix* variant is trained similarly to X-InstructBLIP, with the notable distinction that the modality type is not prepended to the modality's query outputs, both during training and inference. Following (Dai et al., 2023) that noted that sampling ratios play an important role in training we perform some minor modifications in the sampling ratios that we show in tables 13 and **??** are effective in improving performance. The decisions are discussed further below. It is worth noting that due to the large amount of experiments consisting of all modalities, we did not exhaust all possibilities, and there may be better training configurations. We leave this to future work to be explored.

As each modality exhibits unique characteristics, we have customized the training approach for each one. For instance, the 3D and Audio Q-Formers are trained for the maximum number of iterations specified in Table 12.

The Vicuna7b Image Q-Former undergoes training for 735k iterations, utilizing normalized data sampling. Additionally, an extra 40k iterations are performed with the sampling ratio of COCO Captions (Changpinyo et al., 2021) set to 3.0 while keeping the other ratios consistent with the original sampling. This adjustment leverages the clean annotations of COCO Captions, mitigating noise introduced by larger image datasets. However, this upsampling technique is not applied to the Vicuna13b Image Q-Former, since it appears to lower out of distribution performance in non-captioning tasks as shown in table 13. It could be that due to the smaller batch size, Vicuna13b is less sensitive to noisy data, since it effectively sees less of them. In both cases, the last checkpoint from the iterations specified in Table 12 is chosen, with guidance from the COCO Captions validation dataset.

The Vicuna7b video Q-Former is initialized from the best Vicuna7b image Q-Former and undergoes validation every 5k iterations on the MSRVTT captioning (Xu et al., 2016) dataset. The selection process involves choosing the checkpoint that precedes any drop in performance during the subsequent validation rounds even if there is a better performing checkpoint later on in training, to avoid overfitting to the MSRVTT skeletal captions. Table **??** quantitatively shows our observations. Due to the initialization of the video Q-Former with the well trained image Q-Former, the noisy captions

of WebVid2M reduce the performance instead of improving it. However, this is corrected when the training consists of cleaner data.

Similarly, the Vicuna13b video Q-Former is initialized from the best checkpoint of the Vicuna13b Image Q-Former and validated every 2k iterations. While we let the Vicuna7b and 13b video Q-Formers train for 15k and 25k respectively, we observe early convergence at 15k and 4k iterations likely due to the pre-initialization with the Image Q-Former. During training, 5 frames are sampled for the Vicuna7b Video Q-Former, while 4 frames are sampled for the Vicuna13b to reduce computational demands.

The best training approach for each model was empirically identified, and it is beyond the scope of the paper to rigorously analyze the reasons of the differences in training across modalities. We leave this to future work.

| Modality | Model | Iterations | Batch Size |
|---|---|---|---|
| Image | 7b | 775k | 64 |
| | 13b | 880k | 16 |
| Audio | 7b | 65k | 64 |
| | 13b | 300k | 16 |
| 3D | 7b | 65k | 128 |
| | 13b | 300k | 32 |
| Video* | 7b | 5k | 32 |
| | 13b | 25k | 8 |

Table 12: Training hyperparameters. * Video Q-Former is initialized from Image Q-Former.

| | Zero-Shot | | | | | | | In-Domain | |
|---|---|---|---|---|---|---|---|---|---|
| | Flickr30k | NoCaps | | VizWiz | GQA | SNLI-VE | OKVQA | COCO | |
| | test | out-domain | val-all | test-dev | balanced test-dev | test | test | val | test |
| X-InstructBLIP (7b) | 82.6 | 115.5 | 117.5 | 35.5 | 48.1 | 57.6 | 52.5 | 137.1 | 137.9 |
| X-InstructBLIP (7b)-coco | 80.5(↓2.1) | 115.6(↑0.1) | 116.5(↓1.0) | 34.6(↓0.9) | 48.2(↑0.1) | 57.4(↓0.2) | 52.3(↓0.2) | 133.0(↓4.1) | 133.9(↓4.0) |
| X-InstructBLIP (13b) | 77.4 | 114.7 | 115.5 | 34.8 | 49.2 | 58.9 | 52.4 | 130.8 | 130.6 |
| X-InstructBLIP (13b)+coco | 83.6(↑6.2) | 117.2(↑2.5) | 119.3 (↑3.8) | 32.4(↓2.4) | 47.2(↓2.0) | 58.8(↓0.1) | 47.1(↓5.3) | 138.8(↑8.0) | 139.1(↑9.1) |

Table 13: Effect of COCO upsampling.

## F EVALUATION HYPERPARAMETERS

During the evaluation of `X-InstructBLIP`, we adhere to a consistent set of hyperparameters, with minor variations to accommodate the distinct needs of each task. A comprehensive list of these configurations is presented in Table 14. In every experiment, we utilize Beam Search for generation, setting the beam size to 5, repetition penalty and temperature equal to 1.5 and 1 respectively. For tasks involving video-audio modalities, a balanced representation and computational efficiency are achieved by querying two frames from both video and audio. The length penalty is typically configured to 1 for long caption tasks, -1 for Visual Question Answering (VQA) tasks requiring short answers, and 0 for short caption tasks. The minimum and maximum length constraints are adapted based on the task: for captions, we maintain a range of 10 to 80; for short-answer VQA tasks, the range is set from 1 to 10; for variable-length captions, the range is between 1 and 80. In the case of the InstructBLIP baseline for video datasets, we borrow the recommended inference setup of sampling 4 frames for the captioning baselines of MSVD and VATEX with the prompt "A video that shows" and the same generation hyperparameters as `X-InstructBLIP`.

## G INSTRUCTION TUNING SUITE

Table 15 presents a comprehensive list of datasets employed in the instruction tuning process for `X-InstructBLIP`, accompanied by their corresponding dataset sizes. Datasets labeled with ** have been generated automatically through the round-trip-consistency procedure detailed in Section

| Modality | Dataset | Split | Prompt | Len. Penalty | Min Len. | Max Len. |
|---|---|---|---|---|---|---|
| Image | Flickr30k (Van Zwol, 2007) | *test*: 1,000 images | A short description | 1. | 10 | 80 |
| | NoCaps (Agrawal et al., 2019) | *val*: 4,500 images
*out-domain*: 1,413 images | A short description | 1. | 10 | 80 |
| | COCO *(Changpinyo et al., 2021) | *train*: 566,747 image-caption pairs
*val*: 5,000 images
*test*: 5,000 images | A short description. | 1. | 10 | 80 |
| | VizWiz (Bigham et al., 2010) | *test-dev*: 8,000 image-question pairs | based on the given image respond to {question} | -1. | 1 | 10 |
| | OKVQA (Marino et al., 2019) | *test*: 5,046 examples | based on the given image respond to {question} answer | -1. | 1 | 10 |
| | GQA (Hudson & Manning, 2019) | *balanced test-dev*: 12,578 image-question pairs | based on the given image respond to {question} | -1. | 1 | 10 |
| | SNLI-VE (Xie et al., 2018) | *val*: 17,858 image-hypothesis pairs
*test*: 17,901 image-hypothesis pairs | based on the given the image is {hypothesis} true? | -1. | 1 | 10 |
| 3D | Modelnet40 (Wu et al., 2015) | *test*: 2,468 point clouds | Describe the 3d model. A 3d model of | -1. | 1 | 3 |
| | Modelnet40† | | Describe the 3d model. | 0. | 1 | 80 |
| Audio | Clotho (Drossos et al., 2021) | *eval* (v1): 1,045 audios
*val* (v2): 1,045 audios | A short description. | 0. | 10 | 80 |
| | AudioCaps* (Kim et al., 2019) | *train*: 49,838 audio-caption pairs
*val*: 380 audios | A short description | 0. | 1 | 80 |
| Video | MSVD (Chen & Dolan, 2011) | *test*: 670 images[7] | A short description | 1. | 10 | 80 |
| | VATEX(Wang et al., 2020) | *val*: 3,000 images | A short description | 1. | 10 | 80 |
| | MSRVTT *(Xu et al., 2016) | *train*: 130,260 video-caption pairs
*val*: 497 videos
*test*: 2,990 videos | A short description | 1. | 10 | 80 |
| | MSVD QA (Xu et al., 2017) | *test*: 13,157 video-question pairs | based on the given video respond to {question} | -1. | 1 | 10 |
| Audio + Video | AVSD (Alamri et al., 2018) | *dstc7_test*: 6,745 dialogues | {dialog history} {question} | 0. | 10 | 80 |
| | MusicAVQA (Li et al., 2022b) | *test*: 7,402 video-question pairs | Question: {question} Answer: | -1. | 1 | 10 |
| | VALOR (Chen et al., 2023a) | *val*: 2,969 videos | A short description | 1. | 10 | 80 |
| | DisCRn | | Question: {question} Options: first, second Answer: | -1. | 1 | 10 |

Table 14: Hyperparameters used on each of the evaluation datasets. Underlined datasets are in-domain evaluations. * datasets are used for best checkpoint selection. Blue text is provided as input to the LLM but not the Q-Former.

4.1 of the main paper, with further information provided in Appendix A. Datasets marked with •
indicate instances of data loss resulting from file corruption or expired links.

---

[8] Video with id f9_bP219ehQ_63_70 is corrupt and could not be retrieved.

| Modality | Task | Dataset | Training Size |
|---|---|---|---|
| Image | Caption | CapFilt14M (Li et al., 2023b) | 13,873,136 image-caption pairs |
| | | Conceptual Captions 12M (Changpinyo et al., 2021) | 6,029,862 image-caption pairs[•] |
| | | MS COCO Dataset (Lin et al., 2015) | 566,747 image-caption pairs |
| | | SBU Captions (Ordonez et al., 2011) | 859,739 image-caption pairs |
| | | Visual Genome Captions (Krishna et al., 2017) | 821,774 image-caption pairs |
| | QA | AOK VQA (Schwenk et al., 2022) | 17,056 question-answer pairs |
| | | OK VQA (Marino et al., 2019) | 9,009 question-answer pairs |
| | | OCR VQA (Mishra et al., 2019) | 1,002,146 question-answer pairs |
| | | Visual Genome QA (Krishna et al., 2017) | 1,440,069 question-answer pairs |
| | | VQAV2 (Goyal et al., 2017) | 658,104 question-answer pairs |
| | Dialogue | LLaVA150k (Liu et al., 2023) | 394,276 image-instruction pairs |
| Audio | Caption | AudioCaps (Kim et al., 2019) | 38,701 audio-caption pairs[•] |
| | | WAVCaps (Mei et al., 2023) | 297,341 audio-caption pairs[•] |
| | QA | AudioCaps QA** | 24,158 question-answer pairs |
| | Classification | AudioSet balanced train (Gemmeke et al., 2017) | 14,141 labeled audios[•] |
| 3D | Caption | Cap3D (Luo et al., 2023) | 651,576 point cloud-caption pairs |
| | QA | Cap3D QA** | 250,070 question-answer pairs |
| Video | Caption | MSRVTT (Xu et al., 2016) | 130,260 video-caption pairs |
| | | WebVid2M (Bain et al., 2021) | 2M video-caption pairs |
| | QA | MSRVTT QA (Xu et al., 2017) | 149,075 question-answer |

Table 15: Datasets for Instruction Tuning: This table presents datasets used for instruction tuning, along with their associated task types and sizes. [•]Missing data results from expired links and corrupted files. ** Datasets marked with double asterisks are generated automatically within this study.

## H PROMPT TEMPLATES

`X-InstructBLIP` has undergone fine-tuning using a diverse array of instruction templates, tailored to cover a wide spectrum of tasks and modalities. For reference, the specific templates corresponding to each modality can be found in the following tables: Table 17 for images, Table 18 for audio, Table 19 for 3D, and Table 20 for videos. Compared to InstructBLIP (Dai et al., 2023) caption templates have increased from 13 to 32, while question-answering templates have grown from 10 to 21. These enhancements have been strategically incorporated to foster greater adaptability of the model to a wide range of user instructions.

Table 16 presents a performance comparison between InstructBLIP (7b) and `X-InstructBLIP` (7b) on the NoCaps Agrawal et al. (2019) validation set, using prompts not encountered during training. While `X-InstructBLIP` exhibits some performance variability, it maintains a standard deviation that is more than half that of InstructBLIP. This variance can be attributed to the expanded vocabulary in our templates, allowing the Q-Former to better associate an instruction with a specific task. For example, in the case of prompt P2: *Provide a recap of what is happening in the picture"*, InstructBLIP maintains high performance as it closely resembles a prompt seen during training, namely, *Use a few words to illustrate what is happening in the picture"*. It is worth noting that the performance drop in InstructBLIP can be attributed to the language model resorting to generating longer descriptions when the Q-Former outputs have not captured the task, resulting in hallucinations in later stages of generation—a phenomenon also observed in Gunjal et al. (2023).

| | InstructBLIP (7b) | X-InstructBLIP (7b) |
|---|---|---|
| P1 | 1.0 | **88.0** |
| P2 | **121.9** | 109.7 |
| P3 | 0.9 | **54.9** |
| P4 | 5.4 | **112.7** |
| P5 | 0.8 | **111.5** |
| Avg | 26.3 | **83.0** |
| Std. | 43.8 | **20.8** |

| | |
|---|---|
| P1 | In a few words describe the basic features of this image. |
| P2 | Provide a recap of what is happening in the picture. |
| P3 | I'd like to hear your interpretation of this image. What do you see? |
| P4 | Provide a verbal snapshot of what's happening in this image. |
| P5 | Please articulate the elements and context of this image |

Table 16: Robustness to unseen prompts on NoCaps (*vall all*) (Agrawal et al., 2019).

| | **Image Instruction Templates** |
|---|---|
| QA | "{question}"
"Q: {question} A:"
"Answer the following question: {question}"
"Question: {question} Answer:"
"How would you answer {question}?"
"What is the answer to the question {question}?"
"Answer the question based on the image. Question: {question} Answer: "
"Instruction: Answer the following question by reference to the input image. Question: {question} Answer:"
"Given the photo, what is the answer to the question {question}?"
"What's your response to the query {question}?"
"Please provide an answer to {question}"
"Respond to the query {question}"
"Based on the given image, respond to {question}"
"Question: {question} What's your response?"
"Consider the following query: {question}"
"Could you help answer the question {question}?"
"Referencing the provided image, can you answer the question {question}?"
"With respect to the image shown, please answer {question}"
"What's your answer to {question} in the context of the provided image?"
"Question (refer to the image for context): {question} Answer:"
"In response to the question {question}, what would your answer be?" |
| Caption | "A short caption:"
"A short description:"
"A photo of"
"A photo that shows"
"A picture of"
"A picture that shows"
"An image of"
"A image that shows"
"Write a short description."
"Write a description for the image."
"Provide a description of what is presented in the image."
"Briefly describe the content of the image."
"Can you briefly explain what you see in the image?"
"Could you use a few words to describe what you perceive in the image?"
"Please provide a short description of the image."
"Using language, provide a short account of the image."
"Use a few words to illustrate what is happening in the photo." "Write a description for the photo."
"Provide a description of what is presented in the photo."
"Briefly describe the content of the photo."
"Can you briefly explain what you see in the photo?"
"Could you use a few words to describe what you perceive in the photo?"
"Please provide a short description of the picture."
"Using language, provide a short account of the picture."
"Use a few words to illustrate what is happening in the picture."
"Write a description for the picture."
"Provide a description of what is presented in the picture."
"Briefly describe the content of the picture."
"Can you briefly explain what you see in the picture?"
"Could you use a few words to describe what you perceive in the picture?"
"Please provide a short description of the picture."
"Using language, provide a short account of the picture."
"Use a few words to illustrate what is happening in the picture." |

Table 17: Instruction-tuning templates for image tasks

| | Audio Instruction Templates |
|---|---|
| QA | "{question}
"Question: {question} Answer:"
"Q: {question} A:"
"Based on the audio, {question}"
"Answer the following question based on the audio: {question}"
"Question: {question} Provide an answer based on the audio."
"How would you answer {question} based on the audio?"
"What is the answer to the question {question} using the audio as a reference?"
"Answer the question using the audio. Question: {question} Answer: "
"Instruction: Answer the following question by referencing the audio. Question: {question} Answer:"
"Given the audio, what is the answer to the question {question}?"
"What's your response to the query {question} considering the audio?"
"Please provide an answer to {question} using the audio as context."
"Respond to the query {question} based on the audio content."
"Based on the provided audio, respond to {question}"
"Question: {question} What's your response using the audio for context?"
"Consider the following query and the audio: {question}"
"Could you help answer the question {question} using the audio as reference?"
"Referencing the provided audio, can you answer the question {question}?"
"With respect to the audio provided, please answer {question}"
"What's your answer to {question} in the context of the provided audio?"
"Question (refer to the audio for context): {question} Answer:"
"In response to the question {question}, what would your answer be based on the audio?"
"Given the audio, how would you respond to {question}?"
"Taking the audio into consideration, what is your response to {question}?"
"Based on the audio, how would you answer {question}?" |
| Classification | "Classify the following audio:"
"What is the category of this audio clip?"
"Identify the content of the following audio:"
"Provide a classification for the audio."
"Analyze and categorize the following audio."
"Describe the category of the given audio."
"Determine the type of this audio clip."
"Can you classify what you hear in the audio?"
"What type of audio is this?"
"How would you classify this audio clip?"
"Please identify the category of the following audio:"
"What category does the following audio fall into?"
"Classify the sounds in this audio clip." |
| Caption | "A short caption:"
"A short description:"
"An audio of"
"An audio that shows"
"Write a short description."
"Write a description for the audio."
"Provide a description of what is presented in the audio."
"Briefly describe the content of the audio."
"Can you briefly explain what you hear in the audio?"
"Could you use a few words to describe what you perceive in the audio?"
"Please provide a short description of the audio."
"Using language, provide a short account of the audio."
"Use a few words to illustrate what is happening in the audio."
"Describe briefly the contents of the audio."
"Please provide a brief summary of the audio."
"What does the audio contain?"
"What can you hear in the audio?"
"What sounds are present in the audio?"
"Summarize the audio in a few words."
"Write a brief summary of the audio content."
"Could you provide a concise explanation of the audio's contents?"
"Describe what the audio represents."
"What is the audio depicting?"
"In a few words, describe what you hear in the audio." |

Table 18: Instruction-tuning templates for audio tasks

| | 3D Instruction Templates |
|---|---|
| QA | "{question}"
"Question: {question} Answer:"
"Q: {question} A:"
"Based on the 3D model, {question}"
"Answer the following question based on the 3D model: {question}"
"Question: {question} Provide an answer based on the 3D model."
"How would you answer {question} based on the 3D model?"
"What is the answer to the question {question} using the 3D model as a reference?"
"Answer the question using the 3D model. Question: {question} Answer: "
"Instruction: Answer the following question by referencing the 3D model. Question: {question} Answer:"
"Given the 3D model, what is the answer to the question {question}?"
"What's your response to the query {question} considering the 3D model?"
"Please provide an answer to {question} using the 3D model as context."
"Respond to the query {question} based on the 3D model content."
"Based on the provided 3D model, respond to {question}"
"Question: {question} What's your response using the 3D model for context?"
"Consider the following query and the 3D model: {question}"
"Could you help answer the question {question} using the 3D model as reference?"
"Referencing the provided 3D model, can you answer the question {question}?"
"With respect to the 3D model provided, please answer {question}"
"What's your answer to {question} in the context of the provided 3D model?"
"Question (refer to the 3D model for context): {question} Answer:"
"In response to the question {question}, what would your answer be based on the 3D model?"
"Given the 3D model, how would you respond to {question}?"
"Taking the 3D model into consideration, what is your response to {question}?"
"Based on the 3D model, how would you answer {question}?" |
| Caption | "A short caption:"
"A short description:"
"A 3D model of"
"A 3D model that shows"
"Write a short description."
"Write a description for the 3D model."
"Provide a description of what is presented in the 3D model."
"Briefly describe the content of the 3D model."
"Can you briefly explain what you see in the 3D model?"
"Could you use a few words to describe what you perceive in the 3D model?"
"Please provide a short description of the 3D model."
"Using language, provide a short account of the 3D model."
"Use a few words to illustrate what is happening in the 3D model."
"Describe briefly the contents of the 3D model."
"Please provide a brief summary of the 3D model."
"What does the 3D model contain?"
"What can you identify in the 3D model?"
"What structures are present in the 3D model?"
"Summarize the 3D model in a few words."
"Write a brief summary of the 3D model content."
"Could you provide a concise explanation of the 3D model's contents?"
"Describe what the 3D model represents."
"What is the 3D model depicting?"
"In a few words, describe what you see in the 3D model." |

Table 19: Instruction-tuning templates for 3D tasks

| Video Instruction Templates | |
|---|---|
| QA | "Given the video, {question}" |
| | "Q: {question} A:" |
| | "Answer the following question based on the video: {question}" |
| | "Question: {question} Answer:" |
| | "How would you answer {question} after watching the video?" |
| | "What is the answer to the question {question} after viewing the video?" |
| | "Answer the question based on the video. Question: {question} Answer: " |
| | "Instruction: Answer the following question by reference to the input video. Question: {question} Answer:" |
| | "Given the video, what is the answer to the question {question}?" |
| | "What's your response to the query {question} after watching the video?" |
| | "Please provide an answer to {question} after watching the video" |
| | "Respond to the query {question} based on the video" |
| | "Based on the given video, respond to {question}" |
| | "Question: {question} What's your response after watching the video?" |
| | "Consider the following query: {question}" |
| | "Could you help answer the question {question}?" |
| | "Referencing the provided video, can you answer the question {question}?" |
| | "With respect to the video shown, please answer {question}" |
| | "What's your answer to {question} in the context of the provided video?" |
| | "Question (refer to the video for context): {question} Answer:" |
| | "In response to the question {question}, what would your answer be after viewing the video?" |
| Caption | "A short caption for the video:" |
| | "A short description of the video:" |
| | "A video of" |
| | "A video that shows" |
| | "Describe the video briefly." |
| | "Write a description for the video." |
| | "Provide a description of what is presented in the video." |
| | "Briefly describe the content of the video." |
| | "Can you briefly explain what you see in the video?" |
| | "Could you use a few words to describe what you perceive in the video?" |
| | "Please provide a short description of the video." |
| | "Using language, provide a short account of the video." |
| | "Use a few words to illustrate what is happening in the video." |

Table 20: Instruction-tuning templates for audio tasks

