# OpenReview forum: "X-InstructBLIP: A Framework for aligning X-Modal instruction-aware representations to LLMs and Emergent Cross-modal Reasoning"
_ICLR.cc/2024/Conference — ICLR 2024 Conference Withdrawn Submission_

### Official Review · Reviewer_ANX3 · 2023-10-30

**Soundness:** 2 fair
**Presentation:** 3 good
**Contribution:** 2 fair
**Rating:** 5
**Confidence:** 5

**Summary:**

This paper introduces a new multimodal framework based on a frozen large language model, capable of handling and managing various modalities. For training this system, the authors have collected a substantial amount of data involving audio and 3D questions. Additionally, they propose a new evaluation task designed to test the model's performance in cross-modal reasoning.

**Strengths:**

1. The paper is well-written and presents its ideas clearly.
2. A notable contribution of this work is the integration of multiple modalities within the SFT dataset, which furthers the unification of multimodal tasks. This integration is an interesting aspect of the paper.

**Weaknesses:**

1. The paper proposes a multimodal framework built atop a frozen Large Language Model (LLM) aimed at seamlessly integrating and managing various modalities. However, this approach seems to be merely an extension of the existing InstructBLIP.
2. Additionally, the concept of extending to multiple modalities, such as the integration of audio and 3D modalities, has already been proposed in prior works like PandaGPT. Therefore, the paper appears to lack sufficient novelty in both concept and methodology.
3. In Table 1, there is a noticeable drop in performance for X-InstructBLIP. Could you please clarify the reason behind this? If this drop is due to competition among different modalities, do you propose any solutions to mitigate this issue?
4. The promised dataset has not yet been made publicly available, so a cautious approach should be taken regarding this contribution until the dataset is openly accessible.

**Questions:**

Please see weaknesses.

---

### Official Review · Reviewer_SAp4 · 2023-10-31

**Soundness:** 2 fair
**Presentation:** 2 fair
**Contribution:** 2 fair
**Rating:** 5
**Confidence:** 4

**Summary:**

This paper proposes X-InstructBLIP, a multimodal framework based on a LLM, which is able to integrate multiple modalities and use a LLM to accomplish modal fusion. In addition, a new dataset, DisCRn, is proposed to evaluate the model's cross-modal reasoning ability. A large number of experiments have also proved the effectiveness of the method proposed in this paper.

**Strengths:**

- This paper adopts Q-Former and LLM to integrate different modalities, which provides a simple but effective framework and a better method of the fusion between various modalities. The large number of experiments mentioned in the article also prove the effectiveness and potential of this framework
- This paper also proposes a new dataset, DisCRn, to evaluate the model's cross-modal reasoning ability, which fills a gap in related research and provides a reference for subsequent research.

**Weaknesses:**

- The models mentioned in the paper are not given in more detail, e.g. the illustrations about the models should provide information about the encoder of the different modalities
- The framework mentioned in the article can indeed integrate different modalities into LLM, but whether this ability comes from LLM itself or the Q-Former mentioned in the article is not explained in detail. Since it is also common to directly project the embeddings of various modalities into the semantic space of LLM using a projection layer, I don't recognize the advantage of using Q-Former for the model.
- The DisCRn dataset was intended to test model's cross modality reasoning ability, but this dataset mostly evaluates model's reasoning ability in the same modality or in two modalities, and does not involve more modalities simultaneously

**Questions:**

- As the first point I mentioned in the Weaknesses section said, there are few details about the encoder in the paper, so I advise that the authors should add more details and put it in appendix or add an illustration.
- Following up on the previous question, I wonder if the encoders the authors use for different modalities are fixed? If not, I think the effective reasoning ability of your proposed framework comes from the encoders themselves, not from the framework.
- My question about Q-Former and LLM is stated in the second point I mentioned in the Weaknesses section, I wonder the superiority of the whole framework is provided by Q-Former or by LLM itself? It would be great if the authors could provide some relevant experimental proofs!
- The DisCRn dataset only covers the same modality or two modalities, how does the model behave when it receives more modalities? For example, how does the model behave when it receives data input from three or four modalities at the same time?
- In the paper(section 3.2), the authors mention to manually change the sampling rate of MSCOCO and MSRVTT during the training phase,  why this change should be made instead of following the formulas presented in the paper?
- In the experimental section, I noticed that the authors' models with different parameter counts performed inconsistently. Generally speaking, the more parameters a model with the same architecture has, the better the experiment results will be, but I have seen on quite a few tasks that the model with the largest number of parameters (13b) does not work as well as the model with a smaller number of parameters (7b), so I hope that authors could provide an answer to this question

---

### Official Review · Reviewer_3Fuk · 2023-11-01

**Soundness:** 2 fair
**Presentation:** 3 good
**Contribution:** 2 fair
**Rating:** 6
**Confidence:** 3

**Summary:**

The paper introduces a simple but effective multimodal framework with a frozen LLM and  capable of integrating different modality inputs. The authors first collect a high-quality instruction tuning dataset with a  Discriminative Cross-modal Reasoning (DisCRn) evaluation task. Experiments are conducted on 7 zero-shot tasks and the results are comparable to the SOTA.

**Strengths:**

- The paper is well written, easy to follow and understand.
- The intuition is clear, and the results demonstrate most of the the idea works as expected (I do have some concerns on the results, see next section).
- The dataset can be useful to many of the future works.

**Weaknesses:**

- The contribution on the model design is a bit over claiming, as the paper shares very similar idea to the InstructBLIP. From the results, we can also see the paper achieves similar results comparing to InstructBLIP.

- The result section needs more discussion and analysis, for example, the performance gap between the proposed model and other models (tab 1,2) are not well illustrated. Is it because of the data, or the modeling, or something else? Please be specific and discuss instead of listing numbers in the experimental section.

- The paper is about extending the InstructBLIP to multimodality settings, but from the results i dont see the evidence that the multimodality setting here is better than the single modality setting. To be more specific, without the 3D, audio and video modality, will the results in tab1 be much worse? If no, does that mean the proposed multimodality training might harm some of the downstream tasks?

**Questions:**

Please see my weakness section question 2 and 3.

---

### Official Review · Reviewer_fV3B · 2023-11-07

**Soundness:** 2 fair
**Presentation:** 2 fair
**Contribution:** 2 fair
**Rating:** 3
**Confidence:** 4

**Summary:**

This paper proposed X-InstructBLIP for aligning multimodal to LLMs, by extending InstructBLIP to more modalities such as image, audio, video and point clouds. Besides, this paper collect 31K audio QA data and 250K point cloud QA data, and contribute a discriminative cross-modal reasoning evaluation task. The authors evaluate X-InstructBLIP on a set of captioning and question answering tasks across mutlimodal inputs with promising results.

**Strengths:**

- A framework that can support four mdoalities. Previous works can only support one or multiple modalities like image, audio, video and point clouds, while this work unifies them into one model.
- New instuction tuning datasets for audio and 3D point cloud.
- A discriminative cross-modal reasoning evaluation task.

**Weaknesses:**

- The overall contribution is small. X-InstructBLIP is just an extension of InstructBLIP. The only difference X-InstructBLIP utilizes one Q-Former for each modality. However, similar architectures are also explored in X-LLM [1] and CharBridge [2]. The authors contribute new instruction tuning datasets for audio and point clouds, but just an extension of visual instruction data.
- The evaluation results do not demonstrate that X-InstructBLIP is better than pervious works. In Tab.1, the relative improvements compared with previous works are small, even though X-InstructBLIP is trained on there datasets. In Tab.2, we shold note that the results of ChatBridge is zero-shot. So the comparison with CharBridge is unfair. Besides, the MSRVTT result of ChatBridge is wrong. In Tab.4, ChatBridge does not finetune on AVSD and MUSIC AVQA datasets.


[1] Chen, Feilong, et al. "X-llm: Bootstrapping advanced large language models by treating multi-modalities as foreign languages." arXiv preprint arXiv:2305.04160 (2023).
[2] Zhao, Zijia, et al. "Chatbridge: Bridging modalities with large language model as a language catalyst." arXiv preprint arXiv:2305.16103 (2023).

**Questions:**

n/a.

---

### Official Review · Reviewer_hLSA · 2023-11-09

**Soundness:** 2 fair
**Presentation:** 2 fair
**Contribution:** 2 fair
**Rating:** 5
**Confidence:** 4

**Summary:**

This paper focuses on the problem of aligning the LLM models to an ad-hoc number of modalities. To this end, they propose a simple and effective cross-modal framework named X-InstructBLIP for independently aligning multiple modalities to a static LLM. Specifically, they use the Querying Transformer or Q-Former to map inputs from the discrete modality encoder spaces into the language space of LLM. Moreover, they collect high-quality instruction and a new discriminative cross-modal reasoning evaluation. They achieve state-of-the-art benchmarks in seven zero-shot scenarios across all pair modalities.

**Strengths:**

**The motivation is clear and the method is effective.** It is important to unlock the ability of LLM to seamlessly integrate and manage any number of modalities. The proposed Modality-X Q-former is simple and effective.

**The DISCRN evaluation dataset is useful.** It may be useful for evaluation across each of the modalities (images, video, audio, and 3D) alignment.

**Weaknesses:**

**The writing is bad.** It is not clear what is "an ad-hoc number of modalities", do you mean that can extend to any modalities? And in the introduction, it is not clear the differences between your methods with ImagenBind, Imagebind-LLM, and Pandagpt, Can you give me more analysis? In the method section, it is not clear how to train the model with modality-X Q-former, can you provide an algorithm for it?  What about the loss function?

**The proposal method does not have enough technical contribution.** The framework of the methods is more like the BLIP2 with Q-former to more modality. It can not provide more technical insight to the community.

**The experiment is not convincing.** As shown in Table 1 and Table 2 have shown that the X-instructBLIP is not significantly improved with other methods. More importantly, it seems to lack many comparisons, e.g. LLAVA/LLAVA1.5, ImageBind, ImageBind-LLM.

**Questions:**

As shown in weaknesses.